# *Exo*-selective intermolecular Diels–Alder reaction by PyrI4 and AbnU on non-natural substrates

Rajnandani Kashyap [1,2], Naga Veera Yerra[2,3], Joachyutharayalu Oja[4], Sandeepchowdary Bala [1,2], Gal Reddy Potuganti[2,4], Jagadeshwar Reddy Thota[2,3], Manjula Alla [2,4], Debnath Pal[5] & Anthony Addlagatta [1,2✉]

The 100-year-old Diels–Alder reaction (DAr) is an atom economic and elegant organic chemistry transformation combining a 1,3-diene and a dienophile in a [4+2] cycloaddition leading to a set of products with several stereo centres and multiple stereoisomers. Stereoselective [4+2] cycloaddition is a challenge. Here, we describe two natural enzymes, PyrI4 and AbnU performing stereospecific intermolecular DAr on non-natural substrates. AbnU catalyses a single exo-stereoisomer by 32-fold higher than the background. PyrI4 catalyses the same stereoisomer (15-fold higher) as a major component (>50%). Structural, biochemical and fluorescence studies indicate that the dienophile enters first into the β-barrel of the enzymes followed by the 1,3-diene, yielding a stereospecific product. However, if some critical interactions are disrupted to increase the catalytic efficiency, stereoselectivity is compromised. Since it is established that natural enzymes can carry out intermolecular DAr on non-natural substrates, several hundreds of Diels-Alderases available in nature could be explored.

---

[1] Department of Applied Biology, CSIR-Indian Institute of Chemical Technology, Hyderabad, Telangana State 500007, India. [2] Academy of Scientific and Innovative Research (AcSIR), Ghaziabad 201002, India. [3] Analytical and Structural Chemistry Department, CSIR-Indian Institute of Chemical Technology, Hyderabad, Telangana State 500007, India. [4] Fluoro-Agrochemicals Department, CSIR-Indian Institute of Chemical Technology, Hyderabad, Telangana State 500007, India. [5] Department of Computational and Data Sciences, Indian Institute of Science, Bengaluru, India. ✉email: anthony@csiriict.in

The Diels–Alder reaction (DAr)[1] combines a 1,3-diene and an alkene (dienophile) in a [4+2] pericyclic reaction that results in two new C–C bonds and a maximum of four stereocenters[2,3]. DAr has been explored in the biosynthesis of myriad natural products[4]. Several chiral catalysts have been developed to achieve the stereospecificity[5]. In the last three decades, number of enzymes termed as Diels-Alderases (DAses) have been identified that include LovB[6], Sol[7], SpnF[8], PyrE3[9]. PyrI4[10], AbyU[11] AbmU[12] and MalC[13] in the biosynthetic pathways of lovastatin, solanapyrone A, spinosyn A, pyrroindomycins, abyssomicin/neoabyssomicin and fungal alkaloid premalbrancheamide, respectively carrying out intramolecular DAr[14]. Several of these proteins have β-barrel as the active site. Despite little sequence identity (as low as 20%) between these proteins, they share a conserved eight-stranded antiparallel β-barrel core[15] with (+1)8 topology with an α-helix on the outer surface (Supplementary Fig. 1). It has been shown that diene and dienophile are brought together in the β-barrel in an appropriate orientation to provide the required stereochemistry[16,17]. Except for two, all natural DAses reported till date have been demonstrated to carry out intramolecular DAr and only on natural substrates identified in biosynthetic pathways[18]. MaDA and EupfF have been shown to perform intermolecular [4+2] cycloaddition reaction in the biosynthetic pathway of chalcomoracin[19] and tropolonic sesquiterpene[20] related natural products, respectively. To the best of our knowledge, no natural enzyme has been reported to carry out intermolecular DAr with unnatural substrates.

Given the importance of stereospecific DAr in organic chemistry and lack of enzymes that could work on the synthetic substrates, Baker and Houk have pioneered in the design and production of unnatural enzymes[16,21,22] and antibodies[23–25] to carry out the intermolecular DAr[2]. Haptens that mimic the reaction transition state were used to generate antibodies that specifically produce energetically favoured *endo* or disfavoured *exo* Diels–Alder adducts[26]. Baker's group has described the de novo computational design of enzymes catalyzing the intermolecular DAr on synthetic substrates with high stereoselectivity and demonstrated the same experimentally[16]. Using massive computational and molecular biology facilities, of the 84 initial clones, two were found to carry out the DAr. Further mutagenesis led to the discovery of clones that selectively catalyzed *endo* DA product[16,27,28]. Antibodies and artificial enzymes, though could achieve stereospecific DAr on synthetic substrates, their application is limited to very few substrates, specifically to substrates used in the design.

Hydrophobic β-barrel in natural DAses with appropriate placement of hydrogen bond donors and acceptors for stabilization of transition state seems to be critical for the progression of DAr in both natural and designer DAses[16]. There have been no efforts to test if the natural DAses are capable of carrying out DAr on synthetic substrates. We have used PyrI4, a well-characterized natural intramolecular DAse[10] for intermolecular DAr on synthetic substrates. In addition, we have explored AbnU, a hypothetical DAse from *Frankia* sp. Cc1.17 in the possible biosynthesis of neoabyssomicin and a homologue of AbyU (Supplementary Fig. 2)[11,29]. AbnU and the PyrI4 enzymes catalyzed the intermolecular DAr enhancing the rate by 32 and 15-fold higher over the background, specifically favouring the *exo*-stereoisomer.

## Results

### Intermolecular DAr by PyrI4 and AbnU. Genes of AbnU, PyrI4 and PyrI4-Δ10 were synthesized with codon optimization for *E. coli* overexpression and cloned into pET28a (+) vector with N-terminal poly His-tag. E65Q and E65A mutation on PyrI4

backbone and Q115E on PyrI4-Δ10 were achieved by site-directed mutagenesis (Supplementary Table 1). All proteins were expressed, purified to homogeneity and characterized (Supplementary Figs. 3a-d). Protein melting temperatures of PyrI4 and E65Q mutant is 91 °C while its mutants PyrI4-Δ10 and Q115E are 2 °C lower in thermal stability. E65A has gained 3 °C in thermal stability (Supplementary Fig. 3e). Diene (**1a**) and dienophile (**2**) (Fig. 1) were incubated with either AbnU or PyrI4 in phosphate-buffered saline (PBS) solution at 310 K for 2.5 h[26]. A reaction was set up without enzyme under similar conditions to monitor the non-enzymatic DA product formation. The resulted DAr product was subjected to LC-MS and LC-SRM analysis (Supplementary Table 2). Mass spectral data displayed ions at $m/z$ 361.17 and $m/z$ 383.17 which correspond to $[M+H]^+$ and $[M+Na]^+$ ions as expected for **3a** (Supplementary Fig. 4). Intensity of the product was 32 and 15-fold higher in the presence of AbnU and the PyrI4 enzymes respectively, compared to the background reaction (Fig. 2a). This is the first observation of an intermolecular [4+2] cycloaddition reaction catalyzed by any natural enzyme on synthetic substrates.

The kinetic parameters of AbnU and PyrI4 were determined by measuring the dependence of reaction velocity on the concentration of both diene and dienophile (details in the supplementary methods). We observed that diene **1a** binds more tightly (significantly lower $K_M$) to both the enzymes as compared to dienophile **2**. Even though, $k_{cat}$ seems to be much lower for both the enzymes (PyrI4 and AbnU with substrate **1a** and **2**) as compared to the already reported DAses[16,28,30] (with substrate **1b** and **2**, data summarized in Table 1), lower $K_M$ of the substrate and lower $k_{uncat}$, the overall catalytic efficiency is comparable to some of the best evolved artificial enzymes such as CE20. We have compared catalytic efficiency of reported DAses by plotting log (EM) versus log ($1/K_{TS}$) (Fig. 2c).

**Stereo-specificity favouring the *exo*-3S,4S product**. Since the enzymes used in the present study are assisting in the intermolecular DAr on synthetic substrates **1a** and **2**, we investigated into the possible stereoselectivity. Normal phase HPLC was performed using liquid chromatography-mass spectrometry (LC-HRMS) assay with a chiral column (details in supplementary methods)[16,26]. DAr by chemical method using **1a** and **2**, and **1b** and **2** resulted in only four ortho isomers (*exo*-3S,4S, *exo*-3R,4R, *endo*-3S,4R and *endo*-3R,4S) as reported earlier (Supplementary Fig. 5)[16,30]. Note that **1a** is a methyl ester derivative of the acid **1b** (Fig. 1). We used the same analogy as reported earlier for the assignment of stereoisomers of **3a** and **3b**[30]. Only *exo*-3S,4S isomer for the product **3a** was observed when ester **1a** and **2** were incubated with AbnU. On the other hand, a mixture of isomers was observed for **3b** when the acid **1b** and **2** were incubated with AbnU, though *exo*-3S,4S isomer was the major isomer (Fig. 2b, Supplementary Table 3, Supplementary Fig. 6). PyrI4 also favoured the *exo*-3S,4S isomer as a major isomer in catalysing **3a** and **3b**, while the other three isomers were produced in minor quantities. Since **3a** is produced with better stereoselectivity, in the rest of this manuscript, we describe studies with only **1a** diene.

**Crystal structure of PyrI4-Δ10 with Diels–Alder product**. To understand the molecular basis, we determined the co-crystal structure of PyrI4-Δ10 in complex with the Diels–Alder product using X-ray crystallography at a resolution of 2.6 Å (Supplementary Table 4). Since the previous and current structures are no isomorphous, crystal structure of PyrI4-Δ10 was solved using molecular replacement using PDB structure, 5BTU. Dimer of dimer was identified in the asymmetric unit consistent with the

**Fig. 1 Scheme and chemical diagrams of molecules used in this study.** Representative Diels–Alder reaction between 4-carboxybenzyl-trans-1, 3-butadiene-1-carbamate (**1**) and N, N-dimethylacrylamide (**2**) to yield product (**3**). Chemical diagrams of other compounds used in the study are also shown here.

published structure (Fig. 3a)[10]. Efforts to crystallize the full-length PyrI4 with the product were not successful.

Diels–Alder product (**3a**) with all four stereoisomers obtained by synthetic method was used in the co-crystallization experiment (Supplementary Fig. 5)[16]. Clear density was visible for the linear DA product spanning the entire β-barrel in chain A and diffused density in other three molecules of the asymmetric unit (Fig. 3a, b). Further modelling and refinement confirmed that the product bound is *exo*-3S,4S isomer, consistent with the major product obtained with the PyrI4 (Fig. 2b). Though the *exo*-3S,4S is not the major isomer in the synthetic product (Supplementary Fig. 5), selective binding implies that the active site of PyrI4 is suitable to bind substrates such that the resulting transition state and there by the product is linear as in *exo*-isomer better than the U-shaped *endo* product (Supplementary Fig. 7). To establish the identity of the isomers bound in the structure, crystals from the same crystallization drop that was used for X-ray crystallography were washed with mother liquor, extracted with EtOAc and subjected to chiral LC-HRMS analysis (Supplementary Fig. 8a). Both the *exo*-isomers were present with no trace of either of the endo-isomers (Supplementary Fig. 8b). This observation explains the reason for high B-factors for the ligand compared to the surrounding protein residues. The other *exo*-isomer (*exo*-3R,4R) could not be reliably modelled at the current resolution. Several amino acids seem to play a critical role in stabilizing the binding of the product in the β-barrel that includes E65 (Fig. 3c). Both oxygen atoms of the E65 side chain make hydrogen bonds (2.5 Å and 2.6 Å) with the carbonyl of the acrylamide part of the product. This interaction suggests that the E65 carboxylate should be protonated. The dimethyl amino group is buried in the pocket formed by hydrophobic residues (I134, L147, M163, L165) on one side and polar residues (S119 and S121) on the other. Most of the

diene region of the product is buried by Y85, M113, Q115, H117 and Y177 forming hydrophobic contacts. Based on the mode of binding of **3a**, it could be assumed that dienophile (**2**) would first enter the active site followed by the diene (**1a**). It is interesting to note that PyrI4 and SpnF catalyze an *exo*-selective intramolecular DAr on their respective natural substrates[8,10]. Comparison of the PyrI4 structures in complex with **3a** (current structure) and the intramolecular DA product spirotetramate (PDB ID: 5BU3) indicates that **3a** binds deeper in the active site exploring the entire β-barrel while the spirotetramate binds at the entrance of the β-barrel mostly interacting with Q115 (Fig. 3d)[10]. Though the intermolecular and intramolecular substrates explore different regions of the β-barrel of the PyrI4, it is intriguing how the enzyme promotes the *exo*-favoured DAr in both the cases[17].

**Crystal structure of AbnU.** The crystal structure of AbnU in the apo form was determined at 2.0 Å resolution (Supplementary Table 4, Fig. 4a). Efforts to determine the product/substrate bound forms were not successful. Of the several published Diels-Alderases, AbyU is the closest homologue of the AbnU sharing 51.3% sequence identity, while it shares 26% with PyrI4 (Supplementary Fig. 2)[11]. The crystal structure of AbyU (PDB ID: 5DYV) was used to solve the structure. A single molecule is found in the asymmetric unit. It forms a dimer with a 2-fold symmetry-related molecule (Supplementary Fig. 9). Size exclusion chromatography confirms that the protein exists as a dimer in solution (Supplementary Fig. 3c). Similar to the AbyU, the AbnU forms a β-barrel structure with eight β-strands and a short helix insert between the β5 and β6 strands. Analysis of the electron density in the active site suggests that a dimethyl sulfoxide (DMSO) and an ethylene glycol molecule are bound, with the former binding deep

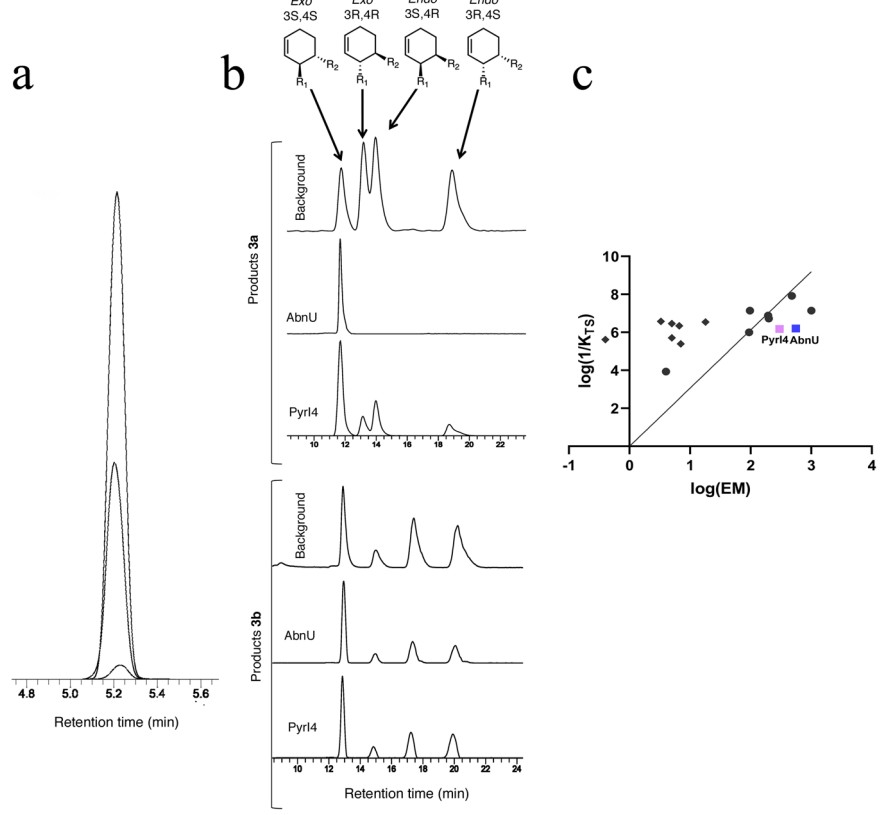

**Fig. 2 Characterization of intermolecular Diels-Alderase reaction by AbnU and PyrI4. a** HPLC analysis of reaction catalyzed by AbnU and PyrI4 and the background reaction. **b** Chiral-HPLC chromatograms of **3a** and **3b** after subtracting the background reaction. All chromatograms without subtraction are shown in the supplementary information (Supplementary Table 5 and Supplementary Fig. 6). Products produced in the non-enzymatic reaction are also shown for the respective reaction as background. Assignment of the stereoisomers is based on the previous literature[30]. On the top chemical diagrams show the relative stereochemistry of each of the stereoisomer. R1 is the diene region of the molecule while R2 is the dienophile part as depicted in Fig. 1. **c** Plot of effective molarity (EM = $k_{cat}/k_{uncat}$) versus catalytic proficiency ($1/K_{TS} = [k_{cat}/K_{M\text{-diene}} \times K_{M\text{-dienophile}})]/k_{uncat}$) for all the reported DAses namely catalytic antibodies and ribozymes (◆), designed artificial DAses (●), AbnU (■, blue) and PyrI4 (■, purple).

**Table 1 Kinetic parameter for AbnU and PyrI4 measured at 37 °C in PBS buffer, pH 7.4.**

| Catalyst | $k_{cat}$ (h⁻¹) | $K_{M\text{-diene}}$, mM | $K_{M\text{-dienophile}}$, mM | $k_{cat}/(K_{M\text{-diene}} \times K_{M\text{-dienophile}})$, M⁻¹M⁻¹S⁻¹ | EM, M | $1/K_{TS}$, M⁻¹ |
|---|---|---|---|---|---|---|
| AbnU | 0.042±0.001 | 0.2±0.01 | 175.1±1.86 | 0.033±0.01 | 557 | $1.6 \times 10^6$ |
| PyrI4 | 0.023±0.002 | 0.2±0.01 | 100±1.25 | 0.031±0.01 | 300 | $1.5 \times 10^6$ |

The estimated errors reflect the SD of three independent measurements. Under similar conditions, $k_{uncat}$ is $0.075 \times 10^{-4}$ M⁻¹h⁻¹. EM represents effective molarity ($k_{cat}/k_{uncat}$)[34]. Chemical proficiency ($1/K_{TS} = k_{cat}/K_{M\text{-diene}} \times K_{M\text{-dienophile}})/k_{uncat}$)[35].

in the pocket (Fig. 4a). We suspect that DMSO could have been trapped in our attempts to co-crystallize the enzyme with the product dissolved in this solvent. Ethylene glycol might be linked to cryo-preservative used to freeze the crystals. Apart from these two molecules, diffused density was observed in the β-barrel possibly for two water molecules. The central β-barrel core formed by the side chains of residues Q12, N15, Q17, Y32, D34, Y51, Y66, F85, V88, Y96 and W114, is mostly hydrophobic (Fig. 4b). A salt bridge formed between E10 and R112 in AbnU seals off the bottom of the β-barrel as seen in AbyU. All polar residues (E10, N15, Q12, Q17 and D34) are aligned on one side of the β-barrel (Fig. 4b). The DMSO molecule is sandwiched between Y32, Y66 and W114 side chains while the ethylene glycol interacts with Q12, Y51 and Y66. The phosphate moiety of the HEPES molecule bound in the AbyU structure[11] and the DMSO in the current AbnU structure are placed in the similar position of the β-barrel making similar contacts (Fig. 4c). The only structural difference between AbyU and AbnU is in the loops connecting

β1-β2 (Loop 1), and β7-β8 (Loop 2). Loop 1 has been implicated in substrate binding in AbyU[11]. The presence of two proline residues (P119 and P122) in loop 2 seem to make it relatively rigid in the AbnU compared to that in AbyU (Fig. 4c).

**Tryptophan fluorescence quenching by 1a, 2 and 3a in AbnU.** W114 is placed deep inside the β-barrel of the AbnU completely exposing its aromatic face to the β-barrel core (Fig. 4b). Tryptophan fluorescence has been invoked to understand several intermolecular interactions that include substrate specificity[31,32]. We have tested the sensitivity of W114 fluorescence in the presence of **1a**, **2** and **3a** to derive mechanistic insights into the DAr in AbnU (Fig. 5a, c, e). Concentration-dependent fluorescence quenching of the Trp was observed in the presence of all three molecules allowing us to calculate their dissociation constant ($K_D$). Surprisingly, DMSO did not perturb the tryptophan fluorescence of AbnU (Supplementary Fig. 10). The dissociation

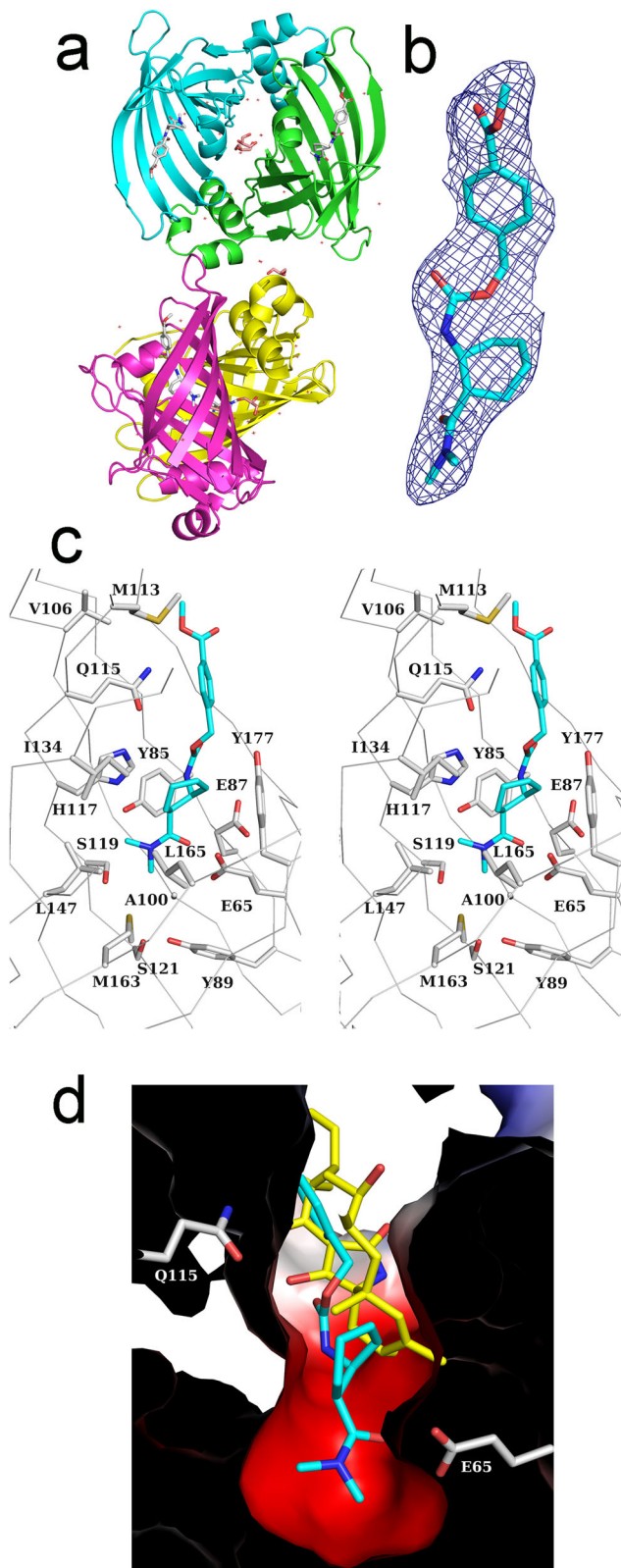

**Fig. 3 Crystal structure of PyrI4-Δ10-3a (*exo*-3S,4S) complex determined at 2.6 Å resolution. a** Cartoon representation of four molecules in the asymmetric unit forming dimer of dimer architecture. Product is shown in white sticks in each of the monomer. **b** 2F$_O$– F$_c$ electron density map (blue mesh contoured at 1.0 σ level) for the *exo*-3S,4S isomer of **3a** in cyan sticks from chain A. **c** Stereo view depicting the detailed interaction of the product (**3a**) with the active site residues of PyrI4-Δ10. Several amino acids make close contact with the product. **d** Electrostatic surface of the enzyme in complex with *exo*-3S,4S product (**3a**) aligned with PyrI4 in complex with the natural product, spirotetramate in yellow sticks (PDB: 5BU3). Compound **3a** binds deeper in the active site of PyrI4 compared to spirotetramate. E65 and Q115 are shown to depict the relative position of the active site.

bind first followed by the diene (**1a**). Therefore, we predict that the dienophile (**2**) would stay longer and probably in a fixed orientation in the β-barrel in the presence of diene (**1a**) than in its absence. This should reflect in the increased W114 fluorescence quenching in the presence of both diene and dienophile, compared to dienophile alone. Fluorescence quenching studies emphatically confirm our hypothesis. At the fixed concentrations of dienophile (**2**) (5 mM) and diene (**1a**) (500 μM), the fluorescence is quenched further compared to dienophile alone (Fig. 5g). On the other hand, fluorescence increase is observed when the dienophile **2** (5 mM) and the product **3a** (500 μM) are incubated (Fig. 5h). Unlike diene, the product competes for the same pocket as dienophile and hence fluorescence quenching is reduced.

**Role of E65, Q115 and the N-terminal lid in PyrI4.** E65 in PyrI4-Δ10 crystal structure makes two critical hydrogen bonds with the product (**3a**) in the dienophile region. We suspect that this interaction efficiently locks the dienophile in a fixed conformation that could promote the *exo*-3S,4S DAr product. Surprisingly, in the case of E65Q mutant, though the overall yield of the product is increased by 50%, stereoselectivity is compromised (Fig. 6a, b). Stereoselectivity is further compromised with E65A mutant providing clues that E65 interactions are critical in supporting the orientation of the dienophile favouring the *exo*-3S,4S product. It is also important to note that in AbnU, E10 is placed structurally equivalent position of E65 in PyrI4 (Fig. 6c). E10 forms a salt bridge with R112. Absolute stereoselectivity and higher yield of **3a** with AbnU could be due to the salt bridge that limits the orientational flexibility and increased activation of the dienophile **2**. The flexible N-terminal lid that includes α0-secondary structure forms a binary complex with the natural substrate of the PyrI4 and has been demonstrated to be critical for the enzyme function[10]. Surprisingly, PyrI4-Δ10 mutant showed 1.8-fold higher activity with the model substrates **1a** and **2** to form **3a** with negligible effect on the stereoselectivity when compared with the full-length enzyme (Figs. 6a, b). Based on the crystal structure of PyrI4-Δ10-**3a** complex, the methyl benzoate part of the product extends out of the β-barrel. In the full-length protein, the R9-D74 salt bridge would cap the **3a** thereby slowing the substrate/product exchange (Fig. 6d). It has been shown earlier by the NMR method that the N-terminus becomes disordered in the PyrI4-Δ10-mutant exposing the core of the β-barrel. It is also important to note that there is no change in the stereochemical outcome in the presence or absence of the lid (Fig. 2a). Q115 makes close contact with the aromatic ring of the product in the PyrI4-Δ10-**3a** structure and was also shown to be crucial for intramolecular DAr in synthesizing the spirotetramate molecule[10]. Though the overall activity of the PyrI4-Δ10-Q115E mutant is higher compared to the wild-type enzyme, preference for stereospecificity of the product is lost signifying that it plays

constant of the product (**3a**) (Fig. 5e, f) is higher compared to the substrate diene (**1a**) (Fig. 5b) suggesting that it may have less effect on the turnover of the catalyst. Higher $K_D$ indicates that small and relatively polar dienophile (**2**) has the least affinity to bind in the β-barrel (Fig. 5c, d). Based on the crystal structure of PyrI4-Δ10—**3a** complex (Fig. 3), dienophile (**2**) is expected to

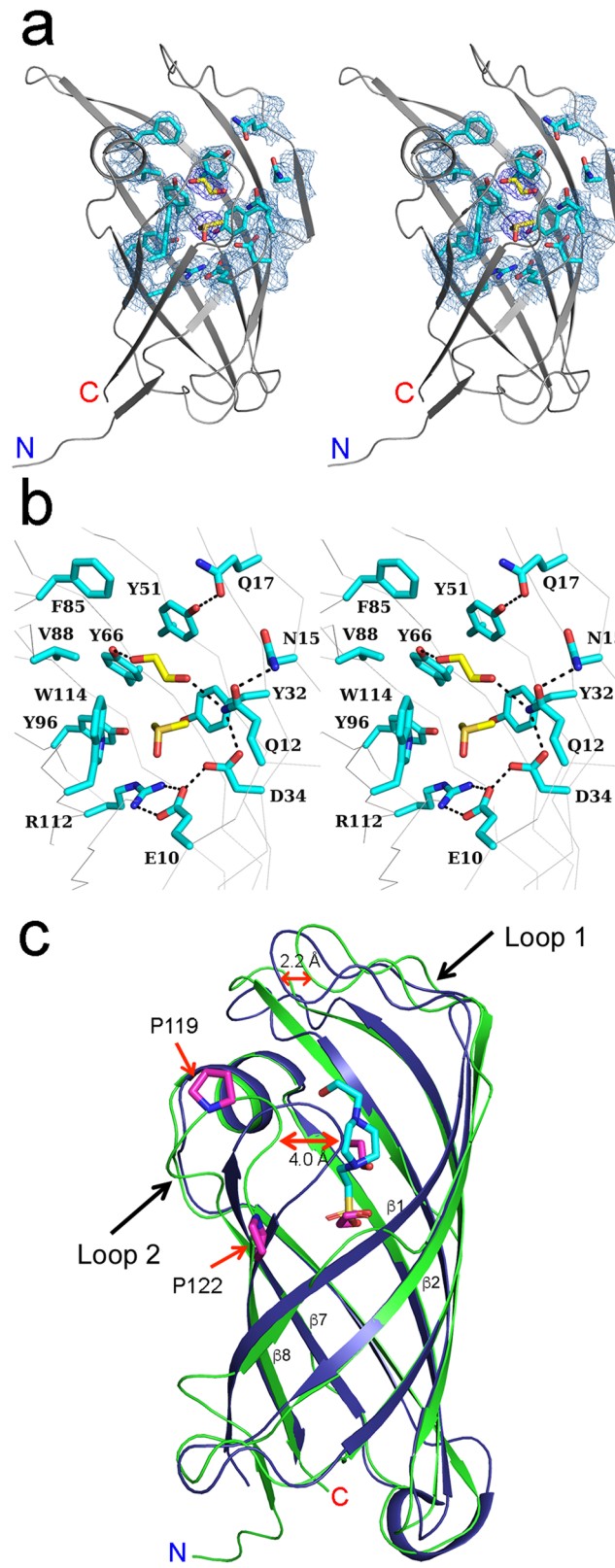

**Fig. 4 Crystal structure of AbnU determined at 2.0 Å resolution. a** Stereo view of the cartoon representation of the AbnU. Active site residues are shown as sticks in cyan covered with 2F$_o$–F$_c$ electron density map (σ = 1.3). An ethylene glycol (EDO) and DMSO molecules are shown as yellow sticks bound in the β-barrel region. **b** Stereo view of AbnU active site depicting the active site residues interacting with EDO and DMSO molecules. Tryptophan (W114) occupies major part of the β-barrel cavity, while E10 and R112 pair-up to form a salt bridge. **c** Cartoon representation of the superposition of AbnU (green) and its homologue AbyU (blue) (PDB: 5DYV). The major difference between the two structures is at the loop region between β1-β2 (Loop 1) and between β7-β8 (Loop 2). The later loop in the AbnU contains two proline (P119 and P122) residues that might restrict the free movement of Loop 2.

(**5**)[16] to test the substrate scope of the wild type PyrI4 and the AbnU. Products formed were analyzed by LC-SRM (Fig. 7, Supplementary Fig. 4). Product **3a**, formed with **1a** and **2** was observed in higher quantities in the presence of both the enzymes. Products **6** and **7** are formed (incubated with **1a** and, **4** and **5**, respectively) in decreasing order of yield compared to **3a**. As the branching increases in N,N-diethylacrylamide, the activity reduces by almost 50%. Surprisingly, in the case of cyclic rigid structure in morpholine-based dienophile (**5**), product (**7**) formed in the presence of PyrI4 is higher than in the presence of AbnU.

## Discussion

DAr is one of the elegant organic reactions discovered to form two new C–C bonds in one go and create a complex molecule from two simple precursors, a 1,3-diene and an alkene. Given its simplicity, DAr applications in the industrial chemistry could have been enormous. In contrast, a limited number of industrial processes have used this reaction[5]. One of the challenges in the practical chemistry of using DAr without the use of an appropriate chiral catalyst is in the separation of mixture of stereoisomers it provides that have the same molecular weight and similar physicochemical properties. Several chiral catalysts were developed to limit the formation of number of stereoisomers[5,33]. Designer Diels-Alderases that include antibodies have provided a single stereoisomer for a given substrate combination raising the hope for the applicability of this reaction in industrial set up[16,26]. However, to the best of our knowledge, enzyme-based methods are yet to be implemented for practical applications. Discovery, biochemical and structural characterization of Diels-Alderases in the biosynthetic pathways of many natural products indicate that the minimum requirement for the DAr to take place is a scaffold that provides cavity and appropriate electrostatics to stabilize the transition state of the substrate(s) for a target reaction[18]. However, none of these natural DAses have ever been tested for their application in intermolecular DAr on synthetic substrates. Given that there exists a scaffold to promote the [4+2] cycloaddition and deep β-barrel, we reasoned that natural DAses could accept two molecules, a diene and a dienophile. PyrI4 is one of the unique DAses with an N-terminal lid crucial for regulating the binding of the substrate in the β-barrel for successful exo-specific intramolecular DAr[10]. We have selected PyrI4 as model enzyme for testing intermolecular DAr with synthetic substrates, expecting the combined support from the shape and electrostatic charge distribution of the β-barrel with the lid playing the regulatory role. In addition, we have selected an uncharacterized hypothetical DAse from Frankia sp. Cc1.17 that is devoid of the N-terminal lid region. We have used substrates reported earlier in the antibody studies[30].

Initial reaction was the incubation of the diene (**1a**) and the dienophile (**2**) with enzyme for 2.5 h at 37 °C and check for the product enhancement over the background. Substantially higher

important role in orienting the incoming diene to prefer the exo-3S,4S product formation (Fig. 6b).

**Substrate scope**. Apart from stereospecificity, substrate diversity also becomes important for any enzyme to find its application. We have used three acrylamide derivatives (N,N-dimethylacrylamide (**2**), N,N-diethylacrylamide (**4**) and 4-acryloyl morpholine

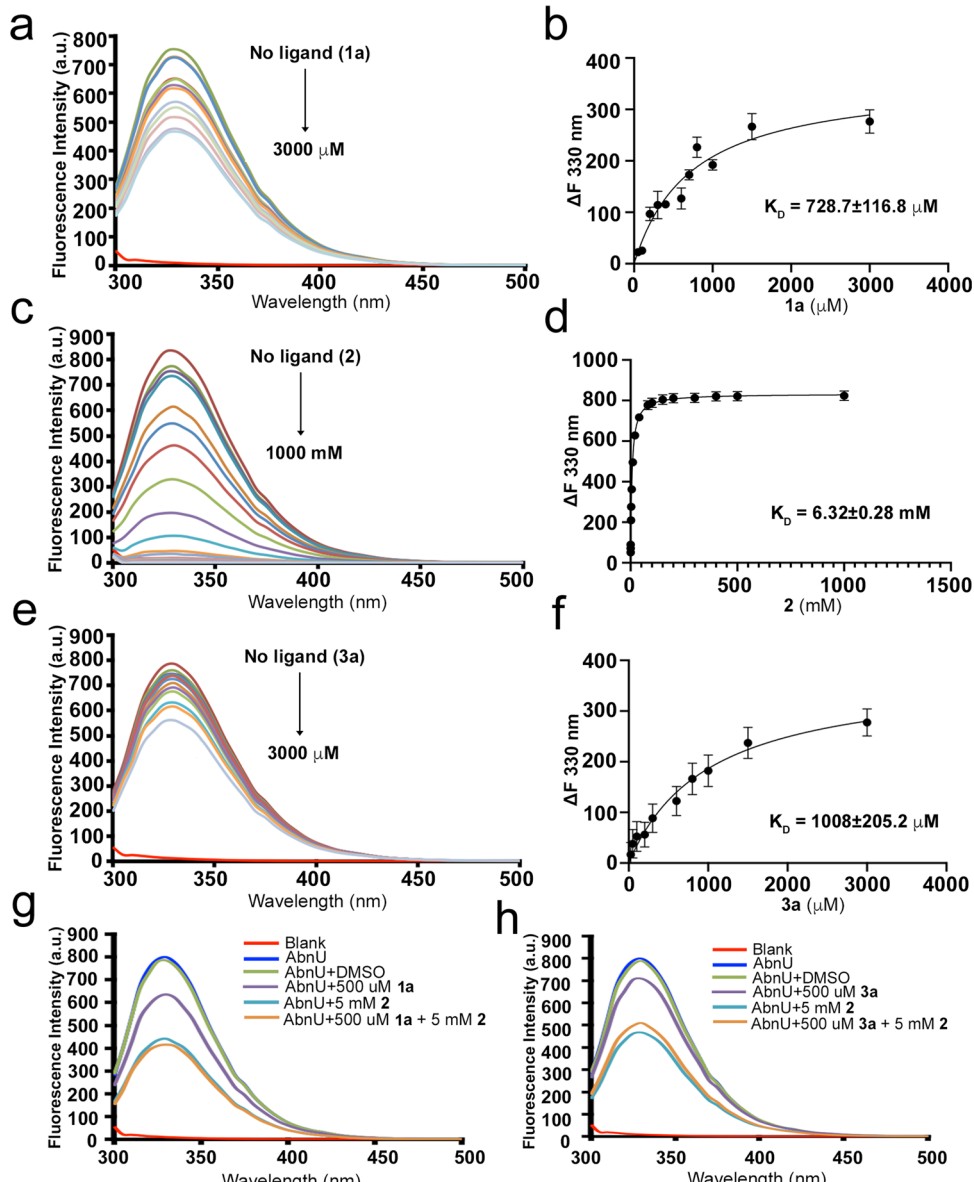

**Fig. 5 Intrinsic tryptophan fluorescence study of AbnU contributed by W114 residue.** AbnU is titrated with increasing concentrations of **1a** (**a**), **2** (**c**) and **3a** (**e**) and the intensity was measured between 300 and 500 nm wavelength. Excitation was carried out at 285 nm. Fluorescence maxima were observed at 330 nm. Fluorescence quenching of tryptophan is noted with increasing concentrations of ligands. (**b**), (**d**) and (**f**) depict the AbnU—ligand affinity ($K_D$) calculated using the one-site-specific binding model[32]. **g** Monitoring of tryptophan fluorescence of AbnU with **1a** (purple), **2** (cyan) and combination of **1a** and **2** (orange). Note that fluorescence quenching is higher in the condition where **1a** and **2** (orange) are present together compared to **2** (cyan) alone. **h** Monitoring of tryptophan fluorescence of AbnU with product **3a** (purple), **2** (cyan) and combination of **3a** and **2** (orange). Note that fluorescence quenching is lower in the condition where the **3a** and **2** (orange) are present together compared to **2** (cyan) alone. It is also important to note that DMSO does not affect the fluorescence of AbnU (green and blue lines in (**g**) and (**h**) panels).

product **3a** was observed over the background reaction (Fig. 2a). AbnU showed absolute stereoselectivity toward the formation of *exo*-3S,4S product. PyrI4 also preferred in catalysing the same *exo*-3S,4S product in higher quantity (51%) along with other stereoisomers (Fig. 2b). We suspect that the E10-R112 salt bridge in AbnU contributes structurally and electrostatically in fixing **2** in single conformation and also activating the alkene bond thereby reducing the LUMO. Crystal structure of the PyrI4-Δ10 in complex with product (**3a**) indicates that the dienophile may enter first into the β-barrel followed by the diene. E65 seem to be critical for orienting and possibly activating the dienophile, while the aromatic and hydrophobic cluster at the other side of the β-barrel position the diene such that the 1,3-diene moiety could

interact with the activated double bond for successful [4+2] cycloaddition reaction. The other evidence for entry of dienophile first comes from the tryptophan fluorescence studies on AbnU. When the diene (**1a**) and the dienophile (**2**) were incubated together, fluorescence quenching is further enhanced, suggesting that diene locks dienophile deeper in the pocket.

The enhanced product formation with AbnU compared to that in the presence of PyrI4 suggests that the lid region in the later this could resist the entry and exit of substrates and product. Also, there is no difference between the stereoselective catalysis with full length and the truncated PyrI4 (Δ10), further conforming that lid region has no role in the intermolecular DAr reported here. Efforts to crystallize the full-length PyrI4 in complex with

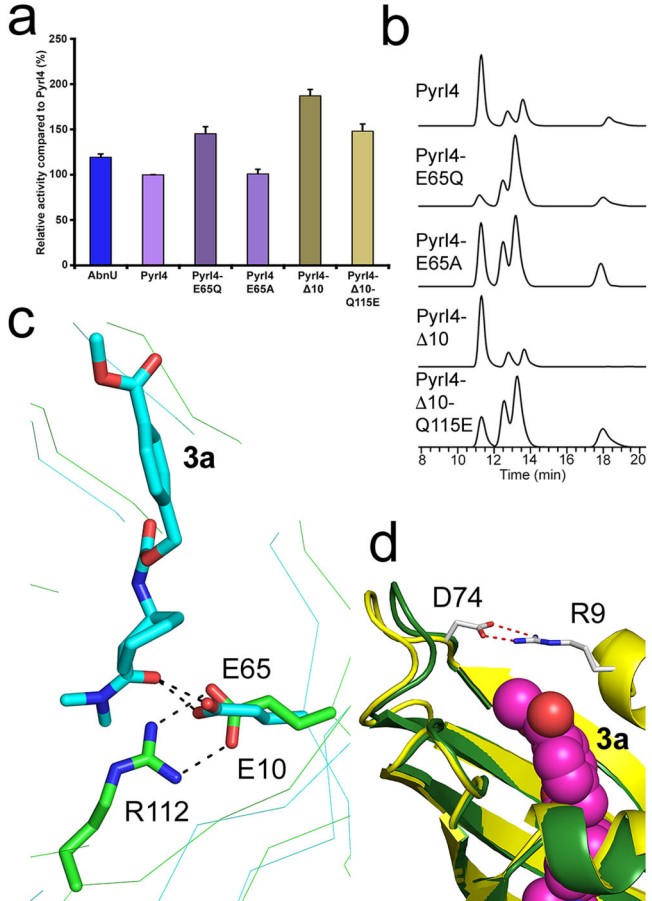

**Fig. 6 Quantification of product, stereoselectivity and intermolecular interactions of PyrI4 and its mutants. a** Quantification of the total product (**3a**) formed when **1a** and **2** were incubated with AbnU, PyrI4 and its mutant by LC-SRM method. Error bars reflect the standard deviation obtained from three independent experiments. **b** Stereoselectivity of the **3a** formed in the presence of PyrI4 and its mutant enzymes analyzed by chiral-HPLC followed by HRMS method. Note that stereoselectivity is similar in catalysing the **3a** from **1a** and **2** with wild type PyrI4 and PyrI4-Δ10. Mutation of E65 and Q115 leads to loss in stereo selectivity. **c** E10 in AbnU (green) is placed structurally equivalent position of E65 in PyrI4 (cyan). E10 forms a salt bridge with R112. Absolute stereoselectivity and higher yield of **3a** with AbnU could be due to the salt bridge that limits the orientational flexibility and increased activation of the dienophile **2**. **d** Superposition of full-length PyrI4 (PDB: 5BU3) and PyrI4-Δ10-**3a** complex showing the salt bridge (R9-D74) at the N-terminal α0 flexible loop capping the product **3a** and possibly slowing the exchange of substrate/product.

the product were not successful, possibly because it interferes with the packing of the lid over the β-barrel. Mutational studies of the PyrI4 indicate that they are important for proper orientation of the substrates for stereoselectivity and not necessarily for the overall yield of the product.

Together, data in this manuscript provides the mechanistic evidence that natural DAses could be employed for intermolecular DAr, specifically on synthetic substrates. Given that there are more than 400 biosynthetic pathways that use DAses, it could be possible to identify the right DAse for a pair of synthetic substrates of choice for industrial application through artificial intelligence. Intuitive engineering of the lead enzyme can further optimize for stereoselectivity, efficiency, stability and reusability, finally finding space in the industrial application.

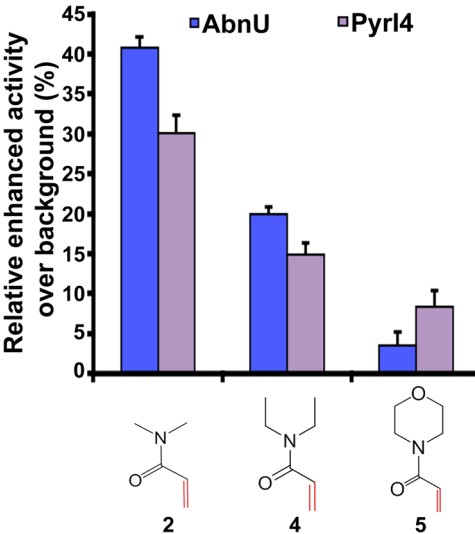

**Fig. 7 Substrate scope of AbnU and PyrI4.** Substrate selectivity of AbnU and PyrI4 for different dienophiles namely **2**, **4** and **5** combined with diene **1a** to yield **3a**, **6** and **7**, respectively. Data shown here represent the enzymatically-enhanced activity over the background reaction. All error bars reflect the standard deviation obtained from independent triplicate experiments.

## Conclusions

In this study, we present first example of natural intermolecular Diels-Alderases carrying out intermolecular DAr reaction on non-natural substrates. We demonstrate that PyrI4, an established intermolecular DAse and AbnU, a homologue of another established DAse, AbyU to carry out intermolecular DAr, with preference to *exo*-stereoisomer. Crystal structure of apo AbnU and PyrI4 in complex with one of the products and, biochemistry and mutagenicity studies provide key mechanistic views about the substrate entry and observed *exo*-stereoselectivity. This study opens a new field of research for exploring natural enzymes in the Diels–Alder chemistry.

## Methods

**Cloning, expression, purification and characterization of proteins**. See Supplementary Methods 1–5, Supplementary Figs. 1–3 and Supplementary Table 1.

**Biocatalysis and product characterization**. See Supplementary Methods 6–9, 13, Supplementary Figs. 4–6, 11 and Supplementary Tables 2, 3.

**Biophysical studies including melting temperatures, fluorescence, X-ray crystallography**. See Supplementary Methods 10–12, Supplementary Figs. 7–10 and Supplementary Table 4.

**Chemical synthesis and characterization**. See Supplementary Method 14, Supplementary schemes 1−3 and Supplementary Figs. 12–19.

**Reporting summary**. Further information on research design is available in the Nature Research Reporting Summary linked to this article.

## Data availability

The data that support the findings of this study are available from the corresponding author on reasonable request. The atomic coordinates and structure factors of PyrI4-Δ10-**3a** and AbnU have been deposited in the protein data bank, www.pdb.org (PDB code: 7DVK and 7DVI). Supplementary information includes the description of all methods, supporting figures and tables with appropriate references.

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

## Acknowledgements

This work was supported by CSIR-IICT internal funds, New Delhi, India. R.K. (DST, India for INSPIRE fellowship), S.C.B. (DBT, India), N.V.Y. and G.R.P. (CSIR, India) acknowledge the respective funding bodies for research fellowships. We acknowledge funding from DST, India for transport of our crystals and data collection at Elettra Synchrotron on beamline XRD2. We thank Dr. Babu Manjashetty, Beamline Scientist, Elettra Synchrotron for collecting data on behalf of us. Thanks to Director, Center for Cellular and Molecular Biology, Hyderabad, India for X-ray data on the home source diffractometer. R.K., N.V.Y., S.C.B. and G.R.P. thank Academy of Scientific and Innovative Research (AcSIR) for Ph.D. registration. CSIR-IICT issued communication number IICT/Pubs./2021/010 for this manuscript.

## Author contributions

The manuscript was written through contributions of all authors. A.A. conceived of and directed the project. R.K. performed molecular biology, biochemical experiments and structural biology studies. N.V.Y. performed mass spectrometry experiments. O.J., S.C.B. and G.R.P. prepared some starting materials for chemistry and molecular biology. J.R.T., M.A. and D.P. have guided the project in the area of mass spectrometry, chemistry and computational work.

## Competing interests

The authors declare no competing interests.
