## [Peer Review File · Communications Chemistry]

Reviewers' comments:

Reviewer #1 (Remarks to the Author):

This paper reports the evolution of 2 natural DAases, PyrI4 and UbnU (at least putative) towards non-natural substrates. Agreeing with the authors, the originality of this work is the evolution of these natural DAases that are known to catalyze intramolecular DA cycloadditions toward intermolecular ones. Another originality is the exo-stereoselectivity of the DA cycloadditions while other examples in the literature describe the endo-selectivity. For these two reasons, this paper is very interesting. It opens new ways in the use of DA-cycloadditions in biocatalysis. However, it suffers of some lacks of important information.

1) The authors prepared some variants E65G/A and Q115E (for Pyr7-D10). What are the rational of these mutations? Why they choosed these mutants? A discussion explaining the choose of these mutations is mandatory.

2) In order to compare with other DAases published in the litterature, kinetic data such as kcat, Catalytic Efficiency (kcat/KMdiene.KMdienophile) and Effective molarity (kcat/kuncat) is also mandatory.

3) This work deserves a better conclusion that the 5 lines at the end the paper.

4) Finally, the recent review of W Ghattas and coll. "Artificial Enzymes for Diels-Alder Reactions" published this year in ChemBioChem 2021, 22, 443 –459 would deserve to be cited.

In absence of these comments and data, I consider this paper not yet suitable for publication.

Reviewer #2 (Remarks to the Author):

This paper has some interesting results, but it creates it own world, divorced from the chemistry known for related reactions. Fact: Diels-Alder reactions are extraordinarily useful, and highly stereoselective catalysts are known. There are also many Diels-Alderases known, but no commercial applications although all authors, including these, of Diels-Alderases always claim that this is why Diels-Alderases are so important!

The authors reported bimolecular Diels-Alder reaction of a benzyl carbamate substituted diene with N,N-dimethylacrylamide catalyzed by biosynthetic Diels-Alderases PyrI4 and AbnU. These substrates were used previously for a designed enzyme by Baker and Houk.

From the abstract, one would have thought that the natural enzymes and substrate were being studied, but actually they just tested several natural enzymes with unnatural substrates. No attempt at measuring acceleration quantitatively was made. The HPLC in Fig. 2a looks like 10-20fold acceleration. The ester a is clearly catalyzed by AbnU, as indicated by stereoselectivity in the HPLC trace, but there is little or no indication of selectivity with the acid, b. These seem to be feeble catalysts, although stereoselectivity is observed with AbuN.

A substrate-enzyme co-crystal structure of PyrI4- Δ 10 was reported. The crystal structure of apo AbnU was reported. These two crystal structures are closely related to their homologues PyrI4 and AbyU, respectively. This is perhaps the most important aspect of this paper.

The catalytic mechanism was postulated by analyzing the product-enzyme complex of PyrI4- Δ 10. Tryptophan fluorescence quenching experiments were conducted to obtain the dissociation

constant of AbnU. The key discovery of this manuscript is that the natural biosynthetic enzymes PyrI4 and AbnU that catalyze DA reaction of two unnatural substrates. One of the authors' conclusions was that more natural DAases could be applied for unnatural substrates for industry applications. Wide substrate scope is important of a method to be applied in industry, but there are only two examples for the reported reaction. While applying synthetic substrates for Diels-Alderase is not new, examples have been reported for catalytic antibody type Diels-Alderase and designer Diels-Alderase. Some of them could even have relatively large substrate scope. It is interesting that natural Diels-Alderase PyrI4 and AbnU catalyze unnatural substrates as the authors reported. It is important to understand the mechanism, but there is lack of mechanistic studies. "The enhanced product formation with AbnU compared to that in the presence of PyrI4 indicate that the lid region in the later is deterrent for entry and exit of substrates and product." This is a guess without mechanistic support.

In the discussion session, the authors made a statement for Diels-Alder reactions that "A major reason for the lack of its applicability in practical chemistry is due to the mixture of stereoisomers it provides that have the same molecular weight and similar physicochemical properties." This is NOT true. First, Diels-Alder reaction has a long history of wide application in chemistry, the basis for many stereoselective syntheses starting with Woodward's reserpine long ago. Many reviews are available on this.

Second, stereochemical control of DA reactions have been extensively studied, used, and explored either controlled by substrate or specific catalyst. "More the number of substituents on the precursors, larger are the number of stereoisomers in the product further complicating the separation process." This statement is not very accurate since increasing number of substituents also affects stereoselectivity and could provide less stereoisomers. "However, to the best of our knowledge, these methods are yet to be implemented for practical applications." Please specify which methods? Again, there are numerous applications of catalyzed stereoselective Diels-Alder reactions. "Given that there are more than 400 biosynthetic pathways that use DAases, it could be possible to identify the right DAase for a pair of substrates of choice through artificial intelligence." I do not agree with this statement. Although there are many biosynthetic pathways involve DAases, without structural details and in-depth studies of the catalytic mechanism, they cannot be properly used.

As the authors stated that this is the first example that natural Diels-Alderase catalyze unnatural substrates. There are now a number of intermolecular Diels-Alderase. Xiaoguang Lei's group in Beijing has hundreds both natural and with unnatural (not all published yet) substrates.

But more in-depth and comprehensive research such as mechanistic studies and substrate scope studies should be performed before publishing. The discussion session should be rewritten. The facts of the paper are interesting, but the claims of general significance and application are overblown, actually false.

A concise discussion of what was actually discovered would be a valuable contribution to the literature.

Minor things:

Page 3. Line 2. Please use "stereocenters" or "stereogenic centers" instead of "chiral centers". The molecule is chiral, the center is stereogenic.

Page 3. Line 13. Please also cite computational studies of PyrI4, J. Am. Chem. Soc. 2020, 142, 47, 20232–20239

Page 3. Line 19. Please change "synthetic substrates" to "unnatural substrates"

Page 3. Line 23. Please add the most recent artificial Diels-Alderase report by Hilvert and Houk.

Page 7. Line 16. Computational studies on PyrI4 have elucidated the origin of its exo selectivity.
Page 10. This entire page is taking about the mechanism of action of PyrI4 which has been clearly elucidated in the previous publication regarding computational studies of PyrI4.

Citations are erratic, for example they should have cited an extensive computational study of PyrI4 by Wen Liu (the discoverer) and Houk: *J. Am. Chem. Soc.* 2020, 142, 47, 20232–20239

Point by point response to the Reviewer's and Editor's comments

Manuscript No.: COMMSCHEM-21-0098-A

Title: Exo-selective intermolecular Diels-Alder reaction by PyrI4 and AbnU on synthetic substrates

Comments from Editor and Reviewers

Reviewer #1: This paper reports the evolution of 2 natural DAases, PyrI4 and UbnU (at least putative) towards non-natural substrates. Agreeing with the authors, the originality of this work is the evolution of these natural DAases that are known to catalyze intramolecular DA cycloadditions toward intermolecular ones. Another originality is the exo-stereoselectivity of the DA cycloadditions while other examples in the literature describe the endo-selectivity. For these two reasons, this paper is very interesting. It opens new ways in the use of DA-cycloadditions in biocatalysis. However, it suffers of some lacks of important information.

1) The authors prepared some variants E65G/A and Q115E (for Pyr7-D10). What are the rational of these mutations? Why they choosed these mutants? A discussion explaining the choose of these mutations is mandatory.

Response: We have discussed the importance of E65 and Q115 in two subsections: 'Crystal structure of PyrI4-Δ10 with Diels-Alder product' and 'Role of E65, Q115 and the N-terminal lid in PyrI4'. In these sections, we described how the E65 interacts with the dienophile region of the product while the Q115 supports the positioning of the diene region with specific hydrogen bonds. Given their importance in holding both the diene and the dienophile, we have reasoned that these two amino acids be mutated as E65Q, E65A and Q115E.

2) In order to compare with other DAases published in the litterature, kinetic data such as kcat, Catalytic Efficiency (kcat/KMdiene.KMdienophile) and Effective molarity (kcat/kuncat) is also mandatory.

Response: We thank the reviewer for this question, which we consider as an important one. To answer this question, we have performed the experiments to determine the catalytic efficiency, effective molarity and catalytic proficiency. Based on these data we have understood that though the catalytic efficiency is low, since the \$K_M\$ of the substrate (1a) is very low for both the enzymes reported in the current

manuscript, the overall catalytic proficiency (\$1/K_{TS}\$ ) of these enzymes is comparable to some of the evolved artificial enzymes like CE20. New data is summarized in Table 1 and Figure 2c.

3) This work deserves a better conclusion that the 5 lines at the end the paper.

Response: We have re-written the conclusions with reference to the results.

4) Finally, the recent review of W Ghattas and coll. “Artificial Enzymes for Diels-Alder Reactions” published this year in ChemBioChem 2021, 22, 443 –459 would deserve to be cited.

In absence of these comments and data, I consider this paper not yet suitable for publication.

Response: We have cited the suggested review article.

Reviewer #2: This paper has some interesting results, but it creates it own world, divorced from the chemistry known for related reactions. Fact: Diels-Alder reactions are extraordinarily useful, and highly stereoselective catalysts are known. There are also many Diels-Alderases known, but no commercial applications although all authors, including these, of Diels-Alderases always claim that this is why Diels-Alderases are so important!

Response: True! We agree with the comment of the reviewer that Diels-Alderases are yet to be used in commercial applications even though they catalyze the formation of two C-C bonds with absolute stereochemistry. In the current manuscript we have tried to explain the reasons for current limitations. We still believe that if a right DAse for a substrate(s) of commercial interest is identified, it could be commercialized. Now that we have shown natural DAases can carryout intermolecular DAr on unnatural substrates with high stereo selectivity, it is possible to explore this class of enzymes in commercial process.

The authors reported bimolecular Diels-Alder reaction of a benzyl carbamate substituted diene with N,N-dimethylacrylamide catalyzed by biosynthetic Diels-Alderases PyrI4 and AbnU. These substrates were used previously for a designed enzyme by Baker and Houk.

From the abstract, one would have thought that the natural enzymes and substrate were being studied, but actually they just tested several natural enzymes with unnatural substrates.

Response: Thank you for pointing this. To bring clarity, we have modified one of the sentences as ‘Here, we describe two natural enzymes, PyrI4 and AbnU performing stereospecific intermolecular DAr on unnatural substrates’.

No attempt at measuring acceleration quantitatively was made. The HPLC in Fig. 2a looks like 10-20fold acceleration. The ester a is clearly catalyzed by AbnU, as indicated by stereoselectivity in the HPLC trace, but there is little or no indication of selectivity with the acid, b. These seem to be feeble catalysts, although stereoselectivity is observed with AbuN.

Response: Thank you for the suggestion. Reviewer 1 posed similar question as well. To answer this question, we have performed the experiments to determine the catalytic and molar efficiency. Based on these data we have understood that though the catalytic efficiency is low, since the \$K_M\$ of substrate is very low for both the enzymes reported in the current manuscript, the chemical proficiency of these enzymes is comparable to some of the evolved artificial enzymes like CE20. New data is summarized in Table 1 and Figure 2c. Yes, there is no selectivity with acid, **1b**. Therefore we decided to perform rest of our studies with the ester, **1a**.

A substrate-enzyme co-crystal structure of PyrI4- Δ 10 was reported. The crystal structure of apo AbnU was reported. These two crystal structures are closely related to their homologues PyrI4 and AbyU, respectively. This is perhaps the most important aspect of this paper.

The catalytic mechanism was postulated by analyzing the product-enzyme complex of PyrI4- Δ 10. Tryptophan fluorescence quenching experiments were conducted to obtain the dissociation constant of AbnU. The key discovery of this manuscript is that the natural biosynthetic enzymes PyrI4 and AbnU that catalyze DA reaction of two unnatural substrates. One of the authors' conclusions was that more natural DAases could be applied for unnatural substrates for industry applications. Wide substrate scope is important of a method to be applied in industry, but there are only two examples for the reported reaction. While applying synthetic substrates for Diels-Alderases is not new, examples have been reported for catalytic antibody type Diels-Alderases and designer Diels-Alderases. Some of them could even have relatively large substrate scope. It is interesting that natural Diels-Alderases PyrI4 and AbnU catalyze unnatural substrates as the authors reported. It is important to understand the mechanism, but there is lack of mechanistic studies. "The enhanced product formation with AbnU compared to that in the presence of PyrI4 indicate that the lid region in the later is deterrent for entry and exit of substrates and product." This is a guess without mechanistic support.

Response: As agreed by the reviewer, our intention is to showcase that natural enzymes can be applied in DA on unnatural substrates. We agree with the reviewer that these two enzymes with the current data cannot be used in commercial chemistry but can provide a clue for exploring natural Diels-Alderases in commercial applications. Our mechanistic studies were focused on the \$\beta\$ -barrel residues that are important

for stereo selectivity. Yes, we did not perform the mechanistic studies on the lid region. The conclusions are drawn based on the biochemical data. We modified the sentence as “The enhanced product formation with AbnU compared to that in the presence of PyrI4 indicate that the lid region in the later could resist the entry and exit of substrates and product”.

In the discussion session, the authors made a statement for Diels-Alder reactions that “A major reason for the lack of its applicability in practical chemistry is due to the mixture of stereoisomers it provides that have the same molecular weight and similar physicochemical properties.” This is NOT true. First, Diels-Alder reaction has a long history of wide application in chemistry, the basis for many stereoselective syntheses starting with Woodward’s reserpine long ago. Many reviews are available on this.

Response: We appreciate the depth of the knowledge of the reviewer in this subject. Our intention in using this sentence was, though it is an elegant reaction in synthesizing complex molecules starting from simple substrates, its application in commercial chemistry is not proportionate. We have replaced the sentence with “One of the challenges in practical chemistry of using DAr without the use of an appropriate chiral catalyst is in the separation of mixture of stereoisomers it provides that have the same molecular weight and similar physicochemical properties”.

Second, stereochemical control of DA reactions have been extensively studied, used, and explored either controlled by substrate or specific catalyst. “More the number of substituents on the precursors, larger are the number of stereoisomers in the product further complicating the separation process.” This statement is not very accurate since increasing number of substituents also affects stereoselectivity and could provide less stereoisomers.

Response: It is our oversight in making this statement. Since this sentence does not have a bearing on the overall discussion, we have removed this sentence.

“However, to the best of our knowledge, these methods are yet to be implemented for practical applications.” Please specify which methods? Again, there are numerous applications of catalyzed stereoselective Diels-Alder reactions.

Response: We have modified the sentence to bring clarity that we are talking about enzyme based methods, “However, to the best of our knowledge, enzyme based methods are yet to be implemented for practical applications”.

“Given that there are more than 400 biosynthetic pathways that use DAs, it could be possible to identify the right Dase for a pair of substrates of choice through artificial intelligence.” I do not agree

with this statement. Although there are many biosynthetic pathways involve DAses, without structural details and in-depth studies of the catalytic mechanism, they cannot be properly used.

Response: Our study has shown that a \$\beta\$ -barrel with appropriately placed residues play important role in catalyzing the DAr. We also show that \$\beta\$ -barrel structure is conserved even though this class of enzymes share very low sequence similarity. Therefore we believe that structures could be modeled for any member of this enzyme class without knowledge of its natural substrates. These models could be used in AI screening for binding of substrates of interest and identify the right enzyme-substrate(s) pair.

As the authors stated that this is the first example that natural Diels-Alderases catalyze unnatural substrates. There are now a number of intermolecular Diels-Alderases. Xiaoguang Lei's group in Beijing has hundreds both natural and with unnatural (not all published yet) substrates.

Response: In this manuscript we claim that AbnU and PyrI4 are natural enzymes performing intermolecular DAr on unnatural synthetic substrates. We have referred the Xiaoguang Lei's paper that describes the intermolecular DAr on natural substrates (*Nat Chem* **12**, 620-628 (2020). Again, to the best of our knowledge, we are not aware of any publication describing any natural enzyme carrying out intermolecular DAr on unnatural substrates.

But more in-depth and comprehensive research such as mechanistic studies and substrate scope studies should be performed before publishing. The discussion session should be rewritten. The facts of the paper are interesting, but the claims of general significance and application are overblown, actually false.

Response: Based on both the reviewers comments and guidance we tried to bring clarity in our discussion. Hope it is acceptable in the current form.

Minor things:

Page 3. Line 2. Please use "stereocenters" or "stereogenic centers" instead of "chiral centers". The molecule is chiral, the center is stereogenic.

Response: I appreciate your suggestion. We have corrected the name from "chiral centers" to "stereocenters".

Page 3. Line 13. Please also cite computational studies of PyrI4, *J. Am. Chem. Soc.* 2020, 142, 47, 20232–20239

Response: Thank you very much for your suggestion. We have cited the suggested article.

Page 3. Line 19. Please change “synthetic substrates” to “unnatural substrates”

Response: We have corrected the name from “synthetic substrates” to “unnatural substrates”.

Page 3. Line 23. Please add the most recent artificial Diels-Alderase report by Hilvert and Houk.

Response: Yes, we have cited this article now.

Page 7. Line 16. Computational studies on PyrI4 have elucidated the origin of its *exo* selectivity.

Page 10. This entire page is taking about the mechanism of action of PyrI4 which has been clearly elucidated in the previous publication regarding computational studies of PyrI4. Citations are erratic, for example they should have cited an extensive computational study of PyrI4 by Wen Liu (the discoverer) and Houk: J. Am. Chem. Soc. 2020, 142, 47, 20232–20239

Response: We have referred this paper in our manuscript at the right place. Liu and Houk made an elegant discussion in their paper regarding the mechanism of *exo*-selective intramolecular product formation by PyrI4. In Figure 3d of our manuscript, we have shown the alignment of intermolecular (current molecule) and intramolecular (spirotetramate) products in the active site of the \$\beta\$ -barrel of PyrI4. It clearly indicates that in the Liu and Houk’s intramolecular mechanism, Q115 is critical for the catalysis and there is no reference of E65. On the other hand we notice that E65 is important in holding the dienophile region of the intermolecular product. Spirotetramate molecule explores only the top part of the \$\beta\$ -barrel, where as the product formed in the current study explores the complete length of the \$\beta\$ -barrel. Hence we had to explore detailed mechanism of intermolecular DAr reported in the current manuscript.

REVIEWERS' COMMENTS:

Reviewer #1 (Remarks to the Author):

The authors have taken into consideration of my remarks and they have answered to my comments. I am satisfied with their answers. As suggested, the authors improved their conclusion that reflects much better their impressive work. I agree with the publication of their article.

Reviewer #2 (Remarks to the Author):

The authors have responded to the reviewers' comments, but indeed, more details reveal the lack of importance of these results. The authors really avoid any mention of the puny two-order of magnitude acceleration; high concentrations would work to speed this up. They make us go to SI or look at a graph and do the calculation for ourselves.

On the other hand they do get selectivity at least with one enzyme, the other may not really be doing any specific catalysis, just changing the medium. The work is technically sound, they obviously put a lot of effort into this, but the claims that now the Diels-Alder reaction will be industrially useful is not correct. In fact, the Diels-Alder reaction is used industrially, many examples are reported in the cited review. The problem is not the reaction efficiency, it is great, but in whether or not the products of a Diels-Alder reaction are needed industrially.

This paper is now technically acceptable. Whether it is sufficiently impactful for Communications Chemistry, with which I am not familiar, I am unable to judge.

Point by point response to the Reviewers' and Editors' comments

Manuscript No.: COMMSCHEM-21-0098-B

Title: Exo-selective intermolecular Diels-Alder reaction by PyrI4 and AbnU on non-natural substrates

Comments from Editor and Reviewers

Reviewer #1: The authors have taken into considerations of my remarks and they have answered to my comments. I am satisfied with their answers. As suggested, the autors improved their conclusion that reflect much better their impressive work. I agree with the publication of their article.

Response: We are glad that we could satisfy you with our responses. Thank you for recommending for publication of our article

Reviewer #2: The authors have responded to the reviewers comments, but indeed, more details reveal the lack of importance of these results. The authors really avoid any mention of the puny two-order of magnitude acceleration; high concentrations would work to speed this up. They make us go to SI or look at a graph and do the calculation for ourselves.

On the other hand they do get selectivity at least with one enzyme, the other may not really be doing any specific catalysis, just changing the medium. The work is technically sound, they obviously put a lot of effort into this, but the claims that now the Diels-Alder reaction will be industrially useful is not correct. In fact, the Diels-Alder reaction is used industrially, many examples are reported in the cited review. The problem is not the reaction efficiency, it is great, but in whether or not the products of a Diels-Alder reaction are needed industrially.

This paper is now technically acceptable. Whether is is sufficiently impactful for Communications Chemistry, with which I am not familiar, I am unable to judge.

Response: We appreciate your comment very much. You have constructively shaped the manuscript. We have now revised the manuscript and included the statement about rate of enhancement clearly in both abstract and introduction. We have also improvised the manuscript and removed the industrial application aspect as per your valuable suggestion. We highly look up to getting our work published in your esteemed journal.